# Applying ICOMOS-IFLA Principles for the Conservation, Management and Reuse of a Historical Hydraulic System: The No-Ras Qanat in North-Western Iran

**Federico Zaina** *, **Paola Branduini** and **Fereshteh Zavvari**

Department of Architecture, Built Environment and Construction Engineering, Politecnico di Milano, P.zza Leonardo Da Vinci 32, 20133 Milano, Italy
* Correspondence: federico.zaina@polimi.it

**Abstract:** Historical hydraulic systems represent a significant part of landscapes and global heritage. However, from the second half of the 20th century onwards, substantial socio-economic as well as technological changes occurring worldwide have put them at risk of abandonment and, eventually, of disappearing. Recent studies and international conventions, including the ICOMOS-IFLA, framed historical hydraulic systems and water management techniques in a new dimension, not only as an element of the past to be preserved but an active element to achieve sustainable economic development and mitigate climate change. Those qanats or karez represented a major historical hydraulic sustainable solution for irrigation, providing a water supply, which during the last few decades, has been slowly replaced with modern, although polluting and unsustainable, technologies. Building on the recent ICOMOS-IFLA Principles Concerning Rural Landscape as Heritage and the recommendation provided by initial research, this paper aims to show how qanats can become: (1) an important local and regional cultural and natural heritage; (2) a valuable economic resource; (3) an environmentally friendly system that could at least partially replace the existing polluting solution (i.e., dams and other modern infrastructures). To achieve these goals, we propose a restoration or reuse approach for the qanat based on the necessity of multiple stakeholders at local and national levels using sustainable materials and respecting the different values as a heritage place. Our case study is the No-Ras qanat in North-western Iran. In the conclusion, we also illustrate the relevance of the aims and methods of this paper in the light of the United Nations Sustainable Development Goals.

**Keywords:** qanat; Iran; historical hydraulic system; landscape heritage; ICOMOS-IFLA; conservation and reuse

## 1. Introduction

### 1.1. Water as Power, Life and Heritage

Water represents the primary human concern and has been one of the major drivers of social, political, cultural and economic development for millennia. The access to water resources contributed to shaping and influencing decision-making from the earliest hunter-gather groups to 21st century countries [1]. Moreover, the first strategies for water manipulation for irrigation purposes, navigation and delimitation of borders emerged as early as the Bronze Age in different regions of the world, including the Near East [2,3], South America [4,5], India [6,7], China [8,9] and Europe [10,11], confirming its paramount significance from small-scale aspects of everyday life up to national scale supply by governmental bodies.

As stressed by Hein et al., water can be a multifaceted tool in support of humans [12], but at the same time, a force against which we should defend when storms or floods break out. For example, the control of river flooding and the regulation of irrigation was one of the first problems that humans had to face following the emergence of agriculture. Over the centuries, the use of water has also generated conflicts between human communities,

such as the political tensions following the construction of dams along rivers crossing more countries in the Middle East [13,14] and central Africa [15,16] or so-called "water grabbing", affecting many African communities [17]. These issues are strictly intertwined with the increasing effects of climate change, representing one of the key challenges in the 21st century. However, the use of water has also led communities around the world to establish an intimate link with it and its benefits. Rivers, lakes and seas have become an integral part of human life and to confirm this deep relationship, stories, traditions and rituals emerged [12,18]. For example, the hydraulic network at Angkor Vat served both the physical infrastructure and the ritual network of sacred places and temples [19]. In the Netherlands, the water management infrastructure represents an iconic element of Dutch historical identity [20]. The key role of water in all aspects of life has also been emphasized in the case of several indigenous Aboriginal Australian communities through the term "cultural water" [21,22]. In other words, water has become heritage.

Today, this can be viewed as both a heritage under threat in many parts of the world due to multiple recent human-induced factors and an active player with a critical role in the mitigation of climate change effects when responsibly used (for example, reconsidering old management systems).

Building on these reflections, this paper aims to investigate a particular type of water heritage: historical hydraulic systems (hereafter HHSs).

### 1.2. Historical Hydraulic Systems as Landscape Heritage

HHSs represent a significant part of rural landscapes and the wider global heritage. According to the geographic location and climate conditions, different types of HHSs have been developed by local communities worldwide. The necessary technical knowledge to build, preserve and manage these HHSs has been passed on for centuries from generation to generation and, over the long term, they have become shared tangible and intangible heritage practices. In addition to being an important cultural asset, HHSs also played a key role in the economy of numerous communities for centuries if not millennia and, in most cases, until a few years ago [23,24].

However, from the second half of the 20th century onwards, substantial socio-economic as well as technological changes occurring at different scales worldwide have put many HHSs around the world at risk of abandonment and eventually of disappearance [25–27]. This issue has slowly come to the fore, thanks to the efforts of international institutions, such as UNESCO (United Nation Educational Scientific and Cultural Organization), ICOMOS (International Council on Monuments and Sites) and IFLA (International Federation of Landscape Architects), as well as numerous researchers worldwide. As a result, a number of conservations and management guidelines for historical landscapes (which include HHSs) emerged during the last few decades, issued by international institutions, including the World Heritage Convention [28], the European Landscape Convention [29] and the Krakow charter [30].

More recently, new studies and international conventions allowed researchers to take a step forward in the understanding of historical landscapes. These have been framed into a new dimension, not only as a passive element of the past to be preserved but also as an active resource to achieve sustainable development and mitigate climate change [31]. Therefore, as stressed by [32], this kind of heritage should not be seen as a burden for the present but as a resource to build a sustainable future and counteract the growing effects of climate change. A systematic definition of the wider role of historical landscapes was provided by the recent ICOMOS-IFLA *Principles Concerning Rural Landscape as Heritage* [33], which recognized them (including HHS) as a resource that can provide food, raw materials and a sense of identity involving economic, environmental, cultural and social aspects. Interpreting HHSs as system of heritage networks connected by functional, physical, social and cultural relationships [34] allows one to overturn previous narratives concerning the passive role of landscape and environment in general, toward an active function in the overall improvement in life quality. Yet, the heritage dimension of HHSs may

also contribute to issues typical of metropolitan areas, including the unregulated urban development and the overall environmental quality. It can offer a number of opportunities for urban populations, such as access to new public spaces, promotion of activities related to the memory and agricultural identity modifying the relationship between rural and urban contexts [32].

The importance of preserving and reusing HHSs has been underlined by several UN Sustainable Development Goals (hereafter SDGs). According to ICOMOS *SDG Policy Guidance* [35], HHSs are an important tangible cultural heritage falling into SDG 11.4 "*strengthening efforts to protect and safeguard the world's cultural and natural heritage*". Moreover, many HHSs are managed through communal initiatives that have been passed down from the past to the present through interactive activities (SDG 4, lifelong learning). Yet, the use of water resources and ingenuous hydraulic systems that are often forgotten and underutilized contributes to SDG 6 (Clean Water and Sanitation) scope and, eventually, to adjust climate variability, thus, matching SDG 13 (Climate action) goals. The effect of the evolution of the concept of historical landscape is also mirrored by the number of calls by the new Horizon Europe program, focusing not only on the preservation, but also on the reuse of natural and cultural heritage as well as on traditional techniques to counteract the effects of climate change [1].

### 1.3. Historical Hydraulic Systems and Climate Change: The Qanats/Kariz

Among the numerous examples of HHSs, a key role is played by the so-called qanat or kariz. This is an ancient system of underground tunnels and wells built for channelling water from a mountain to a generally dry lower region for multiple purposes, including irrigation and drinking water for humans and animals [36–39]. While there is no clear etymological evidence for the qanat or karez (others named it foggara, mayun, negula, etc.), the majority of studies agreed that it might be a Persian or Arabic word meaning "tube", "canal" or "channel" [38–41].

This underground water system represented a major technological solution for water supply in arid and semi-arid regions for millennia [37,39,42], with the earliest archaeological evidence placing its emergence either in the Zagros mountain in the west of Iran [42,43] or southeast Arabia [44,45]. Then, thanks to their multiple social and economic benefits, qanat-like systems spread throughout the Middle East [46] and in many arid and semi-arid regions of China [47,48], the Mediterranean basin [49,50], Northern Africa [42,51] and South America [52].

The specific technical knowledge for construction, management and maintenance gave rise to professional figures called (at least in the Middle East region) *moqanni*, whose skills have often been handed down from father to son for generations [37]. The great engineering and economic value of these works was recognized in ancient times, as confirmed by the Neo-Assyrian chronicles of King Sennacherib [53,54], which brought artisans to Assyria for replicating the system in order to provide water to the main cities of the empire. Entire communities grew and flourished around one or more qanats. Therefore, these HHSs have represented not only an important architectural element and economic engine but also part of the heritage of numerous communities around the world.

Despite this long-term tradition, from the 19th century onwards, an increasing number of qanat/karez have been abandoned and replaced by modern water management systems, such as dams and other hydroelectric infrastructures [36,55,56]. However, recent research brought about a growing consensus on the multiple, often irreversible, issues inherent or caused by most of these modern systems, including pollution and other environmental damage, regional conflicts, political pressures, as well as structural instability [13,14,36,57,58].

Today, qanats/karez are documented in 34 countries around the world [59]. In terms of quantity and dimensions, Iran is the richest country, with almost 24,000 qanats located in the arid and semi-arid dry areas. To help counteracting this phenomenon, over the last decade, a growing number of scientific papers emphasized how the use of HHSs, including qanat/karez, integrated with modern clean and sustainable methodologies and tools,

can become a valuable economic, environmental, social and cultural resource [37,42,60]. Moreover, while detailed studies are still underway, numerous researchers are also stressing the relevance of qanats and similar HHSs to counteract the effects of climate change, often caused by modern hydraulic systems [61,62]. The importance of preserving and reusing these HHSs has been recently confirmed also by UNESCO and FAO (Food and Agriculture Organization of the United Nations). Five UNESCO World Heritage Sites (WHSs) [2] and two FAO Globally Important Agricultural Heritage Systems (GIAHSs) [3] located in Iran or Spain are, or include, one or more qanat/karez.

### 1.4. Applying the ICOMOS-IFLA Guidelines for Documenting, Preserving and Reusing HHSs

The present research builds on a previous paper [36], which analysed the case study of No-Ras qanat, in the Tabriz region of North-west Iran as an example of an HHS at risk of abandonment and decay, that is instead an important cultural and natural heritage (as demonstrated by the UNESCO WHS qanat system of Yazd in central Iran) and that can newly represent a valuable economic resource, also enhancing and reviving the urban layout of a town.

The analysis proposed in that paper regarded the first part, known as "*Principles*", of the guidelines presented by ICOMOS-IFLA [33] and discussed by Scazzosi [34].

The "Principles" phase primarily consists of the definition of the heritage elements and their importance. Moreover, it also takes into account the threats, challenges for conservation as well as the benefit for stakeholders and in terms of a place's sustainability [34]. For our case study, we considered the qanat within its geographic context, considering both tangible and intangible permanencies, the role and involvement of the different stakeholders, the spatial character, the previous and current threats, along with the attitude toward change. We then provided several recommendations, revolving around five key issues, for drafting an efficient action plan, representing the second part of the ICOMOS-IFLA document [33]: 1. water shortage; 2. mismanagement and lack of documentation; 3. loss of technical skills; 4. lack of awareness; 5. perception of qanat as cultural heritage.

This paper focuses on the second part of the ICOMOS-IFLA methodological approach to HHS, meaning the "*Action criteria*". Based on the recommendations proposed in the first part, we will provide an action plan for preserving and reusing the No-Ras qanat.

### 2. Aims and Methods

#### 2.1. Aims of the Research

The wider aim of this article is to contribute to enhancing the debate on the sustainable solutions for water management and the mitigation of climate change effect, based on the conservation and reuse of traditional systems also in urban environments. This will be achieved through a number of specific aims showing how the No-Ras qanat can become: 1. an important local and regional cultural and natural heritage; 2. a valuable economic resource; 3. an environmentally friendly system that could at least partially replace the existing polluting solution (i.e., dam and other modern infrastructures).

These can eventually contribute to meet specific SDGs, in particular, 11.4 (wider aim 1), 4 and 6 (wider aim 2) and 13 (wider aim 3). The wider aims will be achieved by applying the second stage of the methodological approach to rural landscapes, named "*Action criteria*" proposed by the ICOMOS-IFLA *Principles Concerning Rural Landscape as Heritage* [33].

#### 2.2. Methods

The methodology used in this paper starts from the five issues and recommendations provided in Branduini et al. [36] (Table 1). For identifying these issues our previous research, Branduini et al., applied a multidisciplinary approach including landscape architecture, archaeology, remote sensing, engineering and ethnography fields of study [36]. In particular: (1) Landscape architecture and archaeology were used to conduct a thorough territorial and land-use analysis which highlighted the changes in the relation between the Tabriz urbanscape and the surrounding rural area which included also the No-Ras

qanat. (2) Multi-temporal satellite imagery remote sensing was crucial for understanding the long-term damages to the qanat. (3) Engineering provided significant support for the structural analysis of the qanat. (4) Ethnography helped to understand the connection of the local community with the qanat through the perception and awareness analysis.

**Table 1.** Research methodology. Both Issues and Recommendations are from Branduini et al. [36]. The ICOMOS-IFLA Action criteria are taken from ICOMOS-IFLA [33]. RWWTC means Regional Water and Wastewater Treatment Company, RAJO means Regional Agricultural Jihad Organization, ICHHTO means Iranian Cultural Heritage, Handicrafts and Tourism Organization, while MCHT means Ministry of Cultural Heritage and Tourism.

| Issue | Recommendation | ICOMOS-IFLA Action Criteria | Activities Proposed | Actor | Beneficiary |
|---|---|---|---|---|---|
| (1) Mitigate water shortage | Local institutions promoting assessments to compare the cost/benefits of modern hydraulic infrastructure and qanats. If qanats result in a successful water supply, develop a collaborative and sustainable water management plan | **B.3** Defining strategies and actions for dynamic conservation, repair, innovation, adaptive transformation, maintenance, and long-term management **C.5** Finding a balance between long-term conservation and short-term needs | Assessment of cost/benefits of the current hydraulic infrastructures (B.3) Planning and implementing conservation and enhancement activities (B.3) Development of a water management plan (C.5) | RWWTC RAJO University Municipality | Local farmers |
| (2) Mis-management and/or lack of document | Constant monitoring and application of damage assessments by public and/or private research institutes | **A.5** Developing knowledge to enable comparison of rural landscapes at all levels **B.6** Preparing monitoring strategies | Creation of an association/council for managing and monitoring the No-Ras qanat (A.5 and B.6) | RWWTC RAJO Municipality | Local farmers Moqannis and other professionals working in/for the qanat |
| (3) Loss of technical skills | Joint collaboration between local farmers, Regional Agricultural Jihad Organization, and external moqannis for keeping the knowledge | **A.6** Recognizing local populations as knowledge holders **B.4** Consider economic, social, and environmental values **C.2** Recognizing key stakeholders for rural landscapes, including rural inhabitants | Training courses for moqannis and professional figures working in the qanat (A.6 and C.2) Investment in non-invasive sustainable technologies to improve qanat efficiency (B.4) | RWWTC RAJO Local farmers Council of No-Ras qanat | New moqannis and other professionals working in/for the qanat |
| (4) Lack of awareness | Increasing the synergy between universities and local farmers, possibly with the support of the government, in creating projects and associations for the protection and communication of the importance of qanat | **C.4** Considering interconnections between landscapes **D.1** Communicating awareness **D.2** Increasing awareness | Developing signals and panels (D.1-2) Organizing visits to the qanat for children, families, schools, etc. (D.1-2) Organizing conferences and meetings (D.2, C.4) Creating webpages and other online tools (D.1-2) | Universities Local farmers Municipality MCHT Council of No-Ras qanat | Local community |
| (5) Qanat as a heritage | Collecting info about the history of the qanat Applying different communication strategies Involve local communities in conservation activities | **A.4** Inventory and catalogue rural landscapes at all scales **A.7** Promoting cooperation for research **D.3** Supporting shared learning, training, and research | Scientific research on the history of the qanat (A.4, A.7, D.3) Interviews to collect oral history (A.7, D.3) | Universities Local farmers MCHT Council of No-Ras qanat | Local community |

Each issue and recommendation was associated with one or more action criteria listed by the ICOMOS-IFLA guidelines [33].

Based on this, we defined tailor-made activities also identifying actors and beneficiaries. In this way, we aim to move from a traditional top-down approach to a project

co-construction [32] promoting the participation of the public and the government in the mutual elaboration of a landscape project [63]. Moreover, in line with ICOMOS-IFLA guidelines updated interpretation of heritage, we aim to open a discussion on how heritage and its values can help to establish alignments and objectives in terms of economy, social cohesion and environmental protection, creating, in the process, awareness, sense of place and identity in its inhabitants and visitors [60]. A detailed description of the activities, actors and beneficiaries will be provided in Section 3.

### 2.3. Case Study: No-Ras Qanat in Northwestern Iran

We applied this methodology to the No-Ras, located in North-western Iran, on the outskirts of modern Tabriz (Figure 1). It is about 3.5 km long and runs from the slope of Sahand mountain near the village of Chavan to the outlet located in the Fath Abad garden (Figures 2 and 3A).

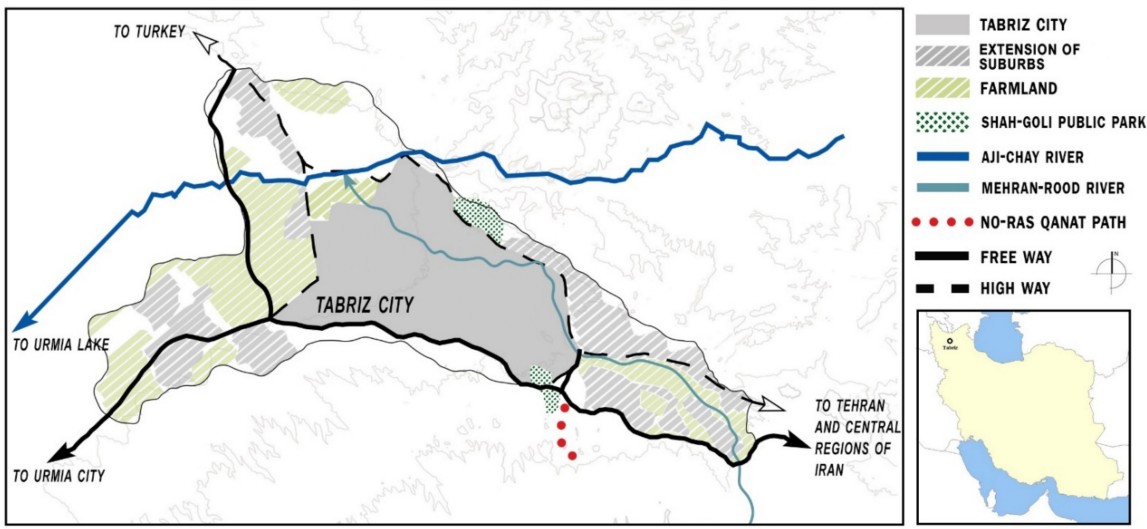

**Figure 1.** The Tabriz area and the No-Ras qanat (after [36]).

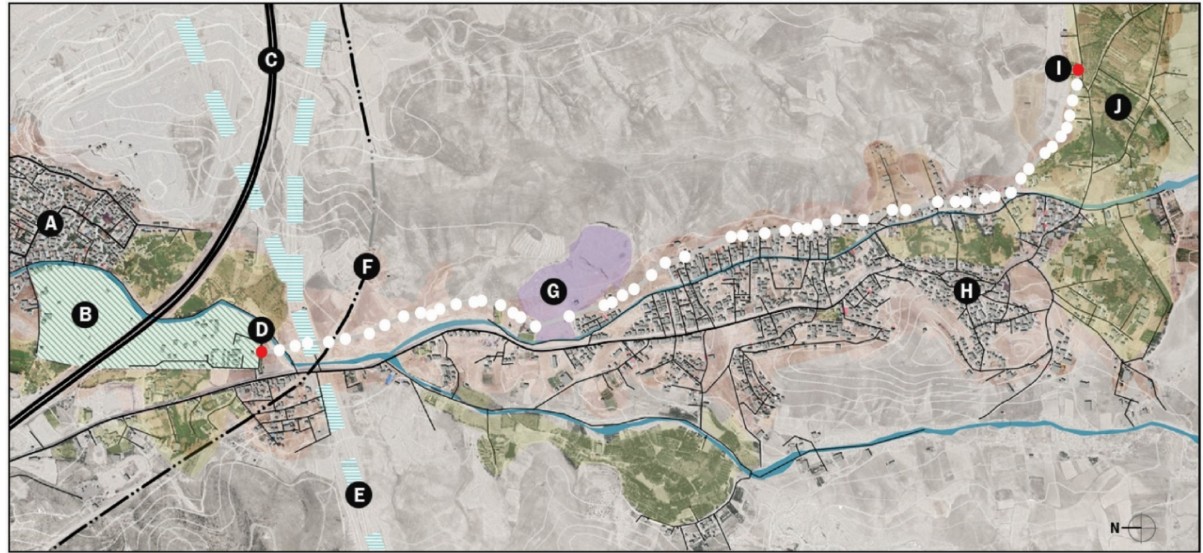

**Figure 2.** The No-Ras qanat; (A) Fath-Abad village; (B) Fath-Abad historical garden (downstream); (C) freeway; (D) outlet/Mazhar (red dot); (E) gas pipeline; (F) train line; (G) sand and gravel mine; (H) Chavan village; (I) mother well (red dot); (J) agricultural lands (upstream).

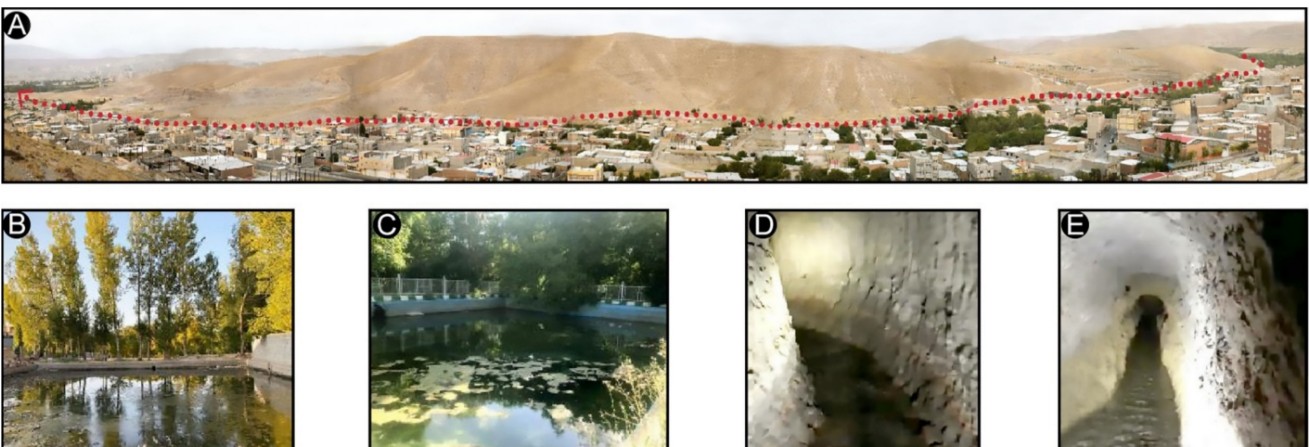

**Figure 3.** The No-Ras qanat. (**A**) Reconstruction of the path in the area of the Chavan village; (**B**,**C**) outlet and pool inside the Fath Abad garden; (**D**,**E**) the underground water system.

The use of the No-Ras qanat is attested as early as the 13th century CE [64], as part of the numerous hydraulic activities carried out by Khajeh Rashid Al-Din Fazlullah during the Mongolian rule over the region. Its construction helped to transform at least part of the dry hilly area south of the city of Tabriz into a fertile land with gardens (i.e., the Fath Abad garden) and cultivated fields. This is also confirmed by Martin et al., who emphasized how the overall plan for the improvement of hydraulic facilities (including the No-Ras qanat) fostered the economic and social development of the region which became a prosperous rural residential area, with facilities and extended farmland [65]. While the political situation in the region remarkably changed through time, this HHS remained in use for almost seven centuries. Today, while at risk of abandonment and slow decay, recent research demonstrated that the No-Ras qanat still holds an important historical value for both the elders of area, the historians and other academic researchers interested in the subject [36]. Moreover, it represents a critical source bringing significant economic benefits to the local farmers and the local authorities, in particular those in charge of the maintenance of the outlet located at the Fath Abad garden [36].

From the pool inside the garden (Figure 3B,C), the underground water system (Figure 3D,E) flows through the Fath Abad village and the surrounding farmland. The No-Ras qanat has 54 wells including the mother well, located at a regular distance of about 60 m. Nowadays, only 11 of these wells can be recognized and just four of them are accessible and open, while the rest are blocked. Several wells have been destroyed by seasonal floods (which also affected part of the gallery) while others were blocked by new owners or fenced within private buildings. The deepest shaft is the so-called "Mother-well" reaching 49 m in depth. Most of the qanat route passes through the village of Chavan and the life and urban development of this small village (counting less than 2000 inhabitants) are strongly interconnected with the (albeit partial) use of the No-Ras. For example, numerous wells are located within the private properties of villagers and many local farmers who exploit the waters of the qanat come from it. Chavan and its inhabitants will, therefore, be carefully considered in the conservation and reuse proposal of the qanat.

The historical relevance of the No-Ras qanat was partially recognized from 1996, when its last stretch, including the Fath Abad garden, the outlet and the historical Garden Mansion were registered in the Iran National Heritage List. This brought to the restoration of the mansion the Fath Abad garden and part of the outlet. However, while the historical mansion was restored according to non-invasive approaches, thus, respecting the architectural layout and the original decorations, both the garden and the outlet were highly modified due to highly invasive methodologies [36]. Further risks, highlighted by Branduini et al., that over the last 20 years increasingly threaten its stability and the possibility of fully reusing it include [36]: buildings and roads construction, soil removal and levelling, ploughing,

pollution and lack of maintenance. These are connected to the five issues illustrated in Table 1 and for which this paper proposes solutions.

## 3. Preserving, Managing and Reusing the No-Ras Qanat

In this section, we describe, in detail, the actors, beneficiaries and the activities that are proposed for the conservation and reuse of the No-Ras qanat, based on the guidelines of the ICOMOS-IFLA document. In fact, although the project has not started yet, numerous contacts with these actors have started and the feasibility of the proposed activities has, so far, been positively evaluated. It is crucial for the design, conservation, management and reuse strategies of the qanat to consider all the stakeholders, their skills, necessities and added value. To facilitate this, Salek [66] recently proposed to apply Actor-Network Theories (ATN) [67,68] to the specific case of qanats to develop a solid network of stakeholders (in our case, defined as actors and beneficiaries). For the scope of our study, ATN stages will be applied to different activities. For example, Sections 3.1 and 3.2, aiming at identifying the actors and beneficiaries, their function, interconnections and necessities in relation to the No-Ras qanat, are consistent with the earliest phases of the ATN, named *Problematization* and *Interessment*.

### 3.1. Actors

Here, we provide the details of the actors already identified during the Principles phase [36] that will play an active role in the individual activities envisaged. Most of them are the same identified by UNESCO and the Iranian Cultural Heritage, Handicrafts and Tourism Organization (ICHHTO) for the national qanats recognized as WHS [69]. For each of them, we specify the role and purpose:

1.  Ministry of Cultural Heritage and Tourism (MCHT): It is the educational and research body overseeing museums, archaeological sites and monuments throughout Iran. One of its branches, ICHHTO, is in charge of managing the qanat at the national and regional levels. Therefore, we believe that the MCHT will be essential for officially supporting the communication and promotion of the cultural value of the qanat at local, national and international scales. In addition to the overall supervision and approval of the activities related to the qanat restoration and reuse, the MCHT will have a primary role in raising awareness of the importance of the qanat (Group of activities 5) and its relevance as heritage (Group of activities 6).

2.  Regional Water and Wastewater Treatment Company (RWWTC): It is a national organization with numerous regional offices in charge of monitoring and developing water and wastewater grids and monitoring the status of urban wastewater treatment systems. They will be involved in restoration and reuse (Group of activities 1), development of efficient management strategies (Group of activities 2) and training (Group of activities 3).

3.  Regional Agricultural Jihad Organization (RAJO): This organization is also in charge of dredging, repairing, reconstructing and developing the Qanat. While their contribution will be crucial over the entire process, together with the RWWTC, they will be mostly involved in restoration and reuse (Group of activities 1), development of efficient management strategies (Group of activities 2) and training (Group of activities 3).

4.  Municipalities: The municipalities of both Tabriz and the village of Chavan represent an important connection between the local communities and the national and regional institutions. They can act as a bridge, facilitating the smooth and clear implementation of the procedures. In our case, based on the recommendations, we believe that they will be important for developing the qanat management protocol (Group of activities 1) for the development of efficient management strategies (Group of activities 2) as well as for contributing to raising awareness of the importance of the qanat (Group of action 4).

5. University: The main research institution of the region is the University of Tabriz. It is one of the most important academic institutions in the country with a wide range of specializations, including agriculture, economy, architecture, engineering and history. Therefore, they can be of the greatest support for finding modern technological sustainable solutions for the qanat restoration (Group of activities 1) as well as for contributing to raising awareness on the importance of the qanat (Group of action 4) and its relevance as heritage (Group of activities 5).

6. Local farmers: These include native people and old immigrants, especially those living in the village of Chavan and Fath Abad, whose main jobs are related to farming, gardening and animal husbandry. They have a good understanding of the value of qanats, which they have extensively used for the irrigation of cultivated lands and gardens as well as for supplying water to their livestock. Their commitment towards the preservation of the qanat is very strong and includes the organization of protests against the heavy damage caused by the sand and gravel mine [36]. Although actively involved in all phases of the project, their qualities will make them important actors to train new professional figures (Group of activities 3) and to raise awareness on the multi-perspective relevance of the qanat (Group of action 4), including heritage (Group of activities 5).

### 3.2. Beneficiaries

Below, we provide details of those stakeholders already identified during the ICOMOS-IFLA *Principles* phase [36], who will benefit from the activities planned. For each beneficiary, we specify the role and purpose:

1. Local farmers: As beneficiaries, they will benefit from all activities related to water shortage mitigation (Group of activities 1), mismanagement (Group of activities 2) and loss of technical skills (Group of activities 3).

2. New moqannis and other practitioners working in/for the qanat: They include the craftsmen specialized in the construction and maintenance of the qanats. The work of the moqanni is usually inherited from father to son or continued among family members [37]. While receiving benefits from all the activities foreseen, they will particularly profit from those related to mismanagement (Group of activities 2) and loss of technical skills (Group of activities 3).

3. Local community: Primarily consisting of the people who live around the Chavan village along the qanat, down to the Tabriz suburbs. Although they will also benefit over the long term from group of activities 1–3, we plan to help them improving their awareness towards the economic benefits and the environmental value of the No-Ras qanat (Group of activities 4), together with the heritage value (Group of activities 5).

### 3.3. Activities

In this section, we detail the single activities listed in Table 1 that are in course of being proposed or approved by the above-mentioned actors. We organized the activities in different groups, each one addressing the issues and recommendations that emerged in the initial research [36]. For example, "Group of activities 1" refers to "Issue and recommendation 1", etc.

#### 3.3.1. Group of Activities 1

The recommendation proposed matches Action criteria B.3 and C.5 in the ICOMOS-IFLA document. Therefore, we designed specific activities to meet their requests. For Action B.3, "*Defining strategies and actions for dynamic conservation, repair, innovation, adaptive transformation, maintenance and long-term management*", we propose two types of activities:

- Assessment of costs and benefits of the current hydraulic infrastructures. This type of analysis must be conducted by the RWWTC, RAJO and the local Municipality in collaboration with the local farmers, who also represent the final beneficiaries. There are currently several approaches to assessing similar situations. Among those, both

cost–benefit analysis (CBA) and cost-effectiveness analysis (CEA) are increasingly applied to archaeological sites, monuments and other heritage places [70,71], while Contingent Valuation (CV) may help in estimating the expected benefits from the qanat for the local residents [72].

- ▪ Planning and implementing preservation and enhancement. The first step regards stopping or modifying all those activities, potentially directly or indirectly, damaging the qanat located within the 15 m buffer zone and, therefore, non-compliant with Iranian law for the protection of cultural heritage as well as the UNESCO regulation for WHS in Iran (Figures 3 and 4). To this end, the qanat area was divided into five sectors, three of which are characterized by a specific type of threat. The upstream sector, defined as "Upstream agricultural fields", does not present immediate risks, although restoration activities of the qanat structure are necessary. Moving towards the valley, the second sector (Figure 4D) corresponds to the northern part of the village of Chavan. Here, 80 buildings are located within the 15 m buffer zone. The majority of buildings include small residential units (22%) or non-used/abandoned ones (30%), while only 7% consist of small-scale industrial complexes. The third sector (Figure 4C) is the one threatened by the mining area, while in the fourth one, the main issues are represented by the railway (Figure 4A) and gas pipeline (Figure 4B) and their debris. While we acknowledge the difficulty of moving a railway and the pipeline, it would be advisable to lift it, thus, making a bridge over the qanat path, so as to substantially reduce the vibrations and pressure caused by trains. The same could be for the pipeline. At the same time, to reduce the number of small vehicles, we suggest an increase in public transport connections between Tabriz and Chavan. This would also facilitate the people living outside the qanat area to easily visit and enjoy it.

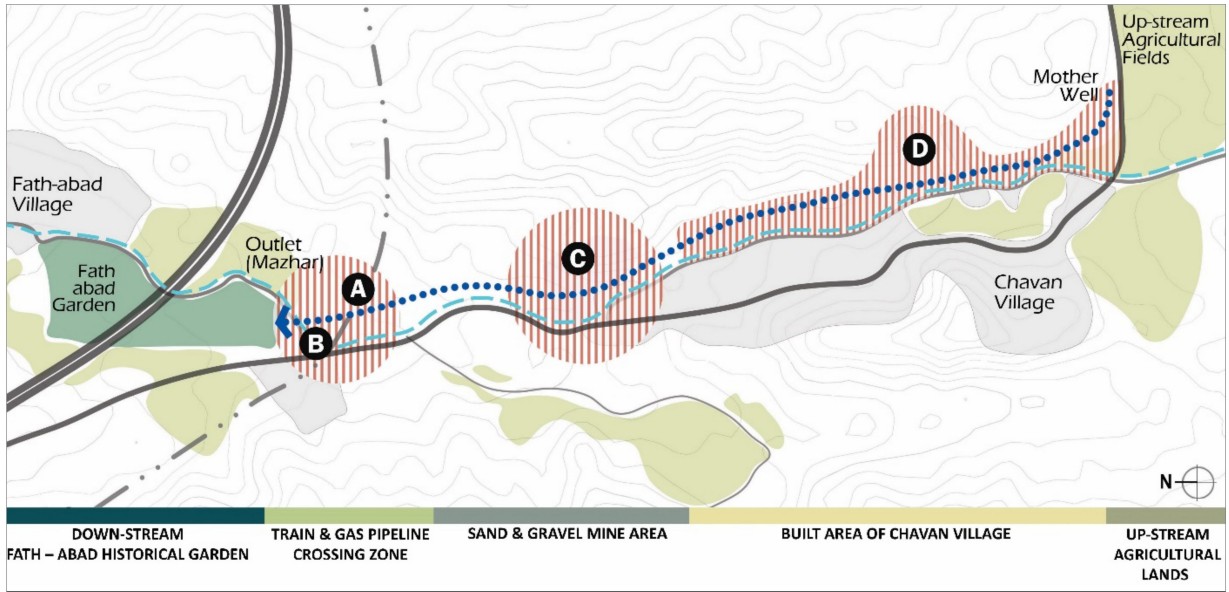

**Figure 4.** Main activities/structures currently threatening the No-Ras qanat stability and use. (A) Railway; (B) gas pipeline debris area; (C) mining area; (D) houses in the Chavan village inside the buffer zone.

The structures affecting the preservation of the No-Ras qanat should be removed and, if possible, relocated elsewhere. Relocation would regard the railway, the pipelines and the houses. Removal without relocation and cleaning would instead regard the mining area and the debris area around the railway line. Mining activities have already demonstrated their dangerous effects on the qanat, causing at least two collapses in the underground gallery and the well in 2008, which caused substantial protests from the local farmers and the Fath Abad authorities.

Once these structures have been removed, No-Ras qanat will be restored. This action will consist in the reopening of the more than 40 wells currently closed and the consolidation of the 11 wells in operation (including the mother well) and the underground channel. The restorations must be carried out using materials compatible (hydraulic mortar, local marl or limestone and black poplar) with the existing ones and that respect the characteristics of the original artifacts.

Once the entire qanat structure is restored and back in operation, we will proceed to the enhancement phase through a series of low-invasive targeted activities in those sectors where previous restoration and enhancement activities have not been carried out by the MCHT or other bodies (i.e., the Fath Abad historical garden).

Transversal Activities

First, the visibility that directly influences the understanding of a qanat will be improved. For this purpose, we envisaged both transversal and targeted activities for the individual sectors. Transversal activities consist of the creation of a boardwalk in selected spots along the underground tunnel, in particular, in the inhabited areas of the Chavan village (Figure 4D) and close to the Fath Abad garden (Figure 4A), located in the outskirts of Tabriz. This is designed to be a low-invasive structure (Figure 5I), mostly composed of local and natural materials, such as black poplar wood, which is also present in the Fath Abad garden. The boardwalk will allow people to walk across, understand and appreciate the multi-faceted landscape around it. To increase the visibility of qanat, next to the wells located along the boardwalk, solar tree panels will be installed together with led lams. The former (Figure 5K) directly catches the sunlight in the morning and generates electricity for lighting in the No-Ras qanat buffer zone, thus, allowing it to be accessible also during the evening.

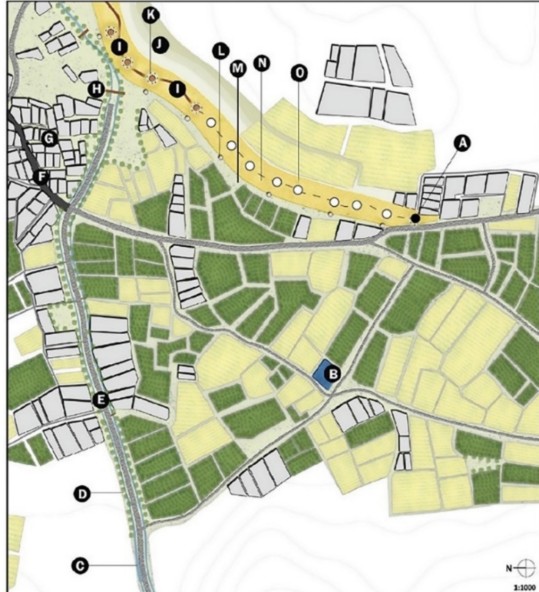
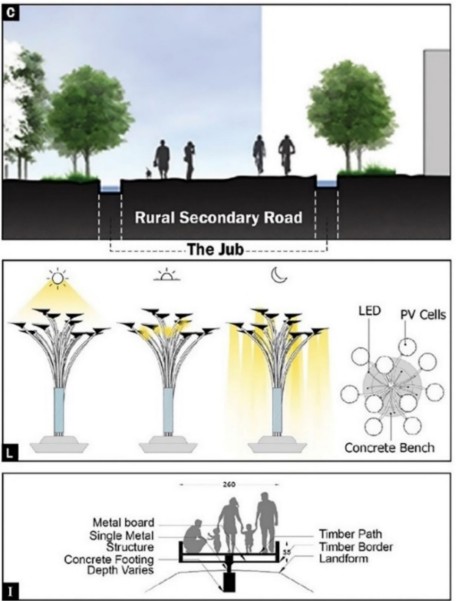

**Figure 5.** Proposal for the renovation of the upstream agricultural area of the No-Ras qanat. (A) Mother well; (B) storage pool for agricultural use; (C) the Jub; (D) newly planted trees; (E) rural secondary road; (F) Chavan main road; (G) Chavan village; (H) small wooden bridge; (I) boardwalk; (J) gentle hill way; (K) No-Ras qanat's wells; (L) information solar tree panels; (M) agricultural fields; (N) No-Ras qanat core zone; (O) limit of the buffer zone.

These renovations will positively impact on at least the northern neighbourhoods of Chavan. Indeed, the more rational the organization and the greater awareness towards the urban (southward) and peri-urban (northward) landscape (Figure 6), the more it will

provide a place of aggregation that, as documented in several cases worldwide [73,74], will foster social inclusion, people's health and possibly the economic value of the houses.

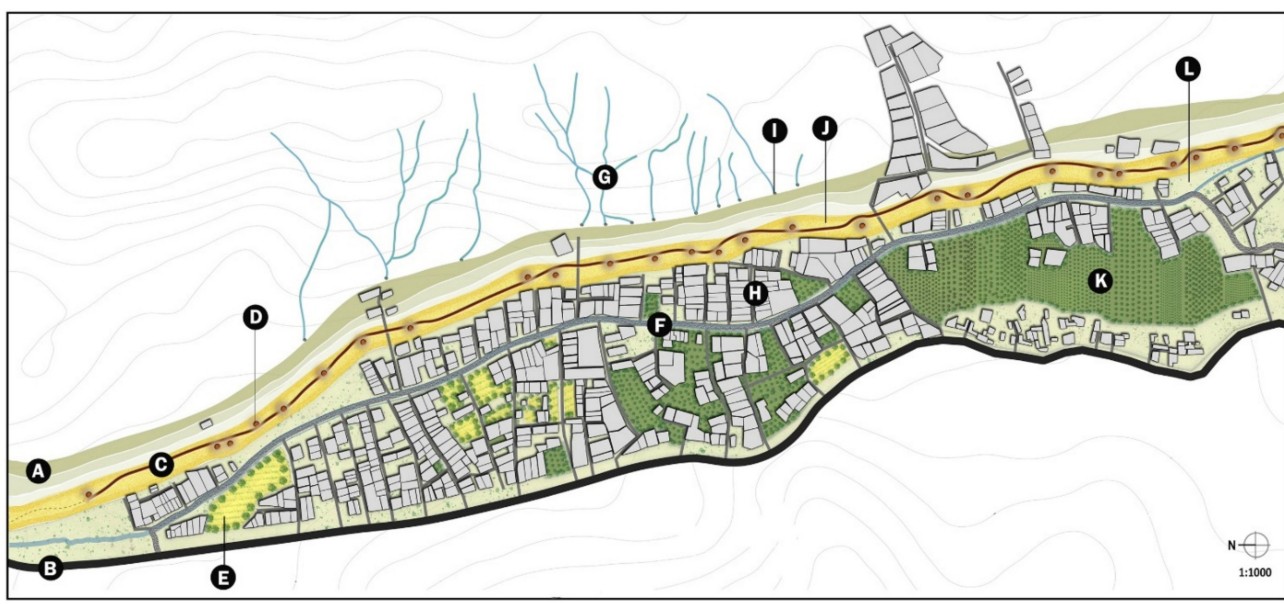

**Figure 6.** Proposal for the renovation of the sector of the No-Ras qanat passing through the modern village of Chavan. (A) Gentle hill way; (B) Chavan main road; (C) boardwalk; (D) No-Ras qanat's wells; (E) centre for organic farming; (F) the Jube; (G) natural seasonal streams; (H) Chavan village; (I) rainwater storage area; (J) No-Ras qanat core zone; (K) reforestation; (L) limit of the buffer zone.

Moreover, in order to foster the understanding of the qanat hydraulic system and the wider natural and the anthropized context in which it is embedded, several information panels will be placed along its path (Figures 5L, 7G and 8I), while numerous wells will be covered with glass slabs to allow people to look inside (Figures 7C and 8B).

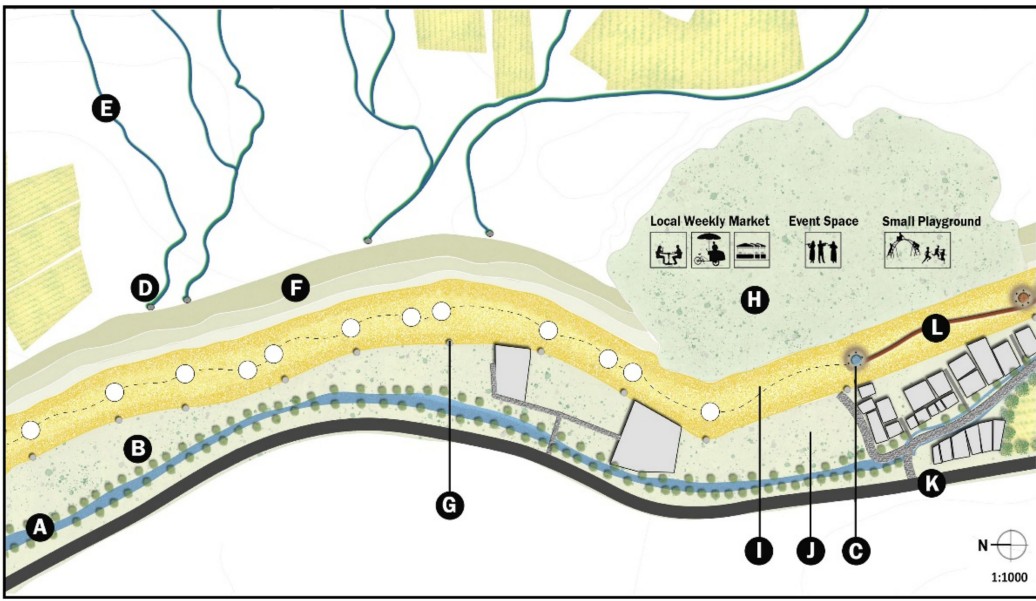

**Figure 7.** Proposal for the renovation of the mining sector of the No-Ras qanat. (A) River; (B) newly planted trees; (C) No-Ras qanat's wells (Glass slabs); (D) rainwater storage area; (E) natural seasonal stream; (F) hill; (G) information solar tree panels; (H) new public square for the local market; (I) No-Ras qanat core zone; (J) open areas (No-Ras qanat buffer zone); (K) Chavan main road; (L) boardwalk.

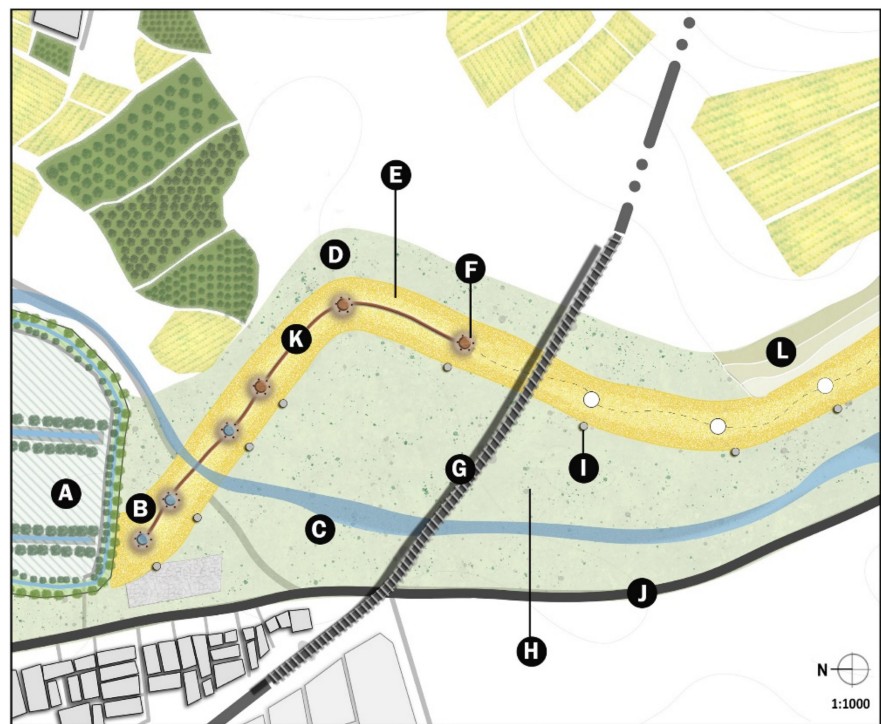

**Figure 8.** Proposal for the renovation of the railway and pipeline sector of the No-Ras qanat. (A) Fath Abad historical garden; (B) No-Ras qanat's wells (glass slabs); (C) river; (D) No-Ras qanat buffer zone; (E) No-Ras qanat core zone; (F) No-Ras qanat's wells; (G) new railway and gas pipeline bridge; (H) open areas; (I) information solar tree panels; (J); Chavan main road; (K) boardwalk; (L) gentle hill way.

A small seasonal river passes through the village of Chavan, flowing close to the No-Ras qanat. This river stream, running both in the upstream area and the successive ones, corresponds to the path of the secondary road (Figure 5E), which during the fall and winter seasons, is partially or fully flooded. Moreover, when running close to the qanat, it may affect its stability and access to the wells. Therefore, we propose a solution for controlling the water flow and improving the area seasonally flooded by the river. We suggest using a traditional water management system called "Jube" (Figures 5E and 6F) [75]. This consists of the division of the river into two separate streams located on both sides of the secondary road. The outer banks of the river will be strengthened by planting local trees, while the road will be kept unpaved, meaning made of the local soil.

As a traditional hydraulic system and part of the heritage of the region, the Jube will be critical for managing and preventing new floods in the northern part of the city. Moreover, the new layout will provide an important and safe roadway crossing the entire village and it will contribute to the embellishment of the area, possibly leading, over the long term, to an increase in the economic value of the buildings along it.

Target Activities for the Upstream Agricultural Area

Targeted activities regarding the "Upstream agricultural area" include the creation of a large pool for agricultural purposes, representing a traditional solution in the region for storing water (Figure 5B). Similar infrastructures are currently missing in the area and their creation would be a valuable support for local farmers, especially in dry seasons.

Target Activities for the Sector Occupied by the Chavan Village

In addition to the transversal action described above, two main activities have been planned to enhance the sector occupied by the Chavan village (Figure 6).

A first structural action, also affecting the sector occupied by the mining area, aims at protecting the qanat from the natural seasonal streams flowing from the top of the hill to

the east (Figure 6G). The erosive action of the seasonal streams causes substantial damage to the qanat over time. To avoid this, we propose placing several small wells along the streams to allow for natural storage of the water. In this way, we will also provide farmers with additional water for irrigation.

A second initiative concerns the creation of a "Centre for organic farming" (Figure 6E). This center will support farmers in the development of sustainable solutions for agriculture, thus, contributing to reducing pollution generated by intensive agriculture and eventually meeting the scopes of SDG 13. The center will involve the local population through events and lectures, during which the key concepts of organic farming and their benefit for both people and the environment will be explained.

The creation of the Centre of organic farming could be supported by national funding as well as from the municipalities of Chavan and Tabriz. Scientific and technical supports may also be searched among international organizations and NGOs, including the Organic Research Centre (https://www.organicresearchcentre.com/ (accessed on 11 October 2022)) and the International Society of Organic Agriculture Research (https://www.isofar.online/Home/ (accessed on 11 October 2022)). In addition to the provision of new job positions, the novel approach of the center (i.e., promotion of organic farming) may help attracting researchers, external farmers and other stakeholders, thus, making the village an important hub for organic farming in the area. Over the long term, the center may host fairs and events, also attracting people from the entire region or nation, thus, having economic benefits for the population of Chavan and Tabriz as well.

Target Activities for the Sector Occupied by the Mining Area

In addition to the transversal activities, we propose replacing the existing mining area with a public square hosting a weekly market (Figure 7H). The square will be equipped with solar tree panels and soft park furniture lighting. The weekly markets will be organized in collaboration with the Centre for organic farming to sell the products of local farmers, promote sustainable agriculture and share the value and results of the No-Ras qanat renewal project.

A second action envisaged in the mining area will regard the reforestation of a large portion, located (Figure 6K) close to the Chavan village with local trees. This action will have a significant environmental impact, eventually contributing to meet the scopes of SDG 13. As demonstrated by numerous studies worldwide [76–79], the reforestation of the areas around villages and towns have a positive impact on the urban population, also increasing the value of buildings and possibly triggering a virtuous circle of urban improvement.

Target Activities for the Sector Occupied by the Railway and Pipeline

Most of the changes proposed for the railway and gas pipeline area (Figure 8G) fall within the transversal activities (see above). Targeted activities for this sector include the creation of two large open areas with meadows and local plants (Figure 8H) to the north and south of the qanat buffer zone. In addition to revitalizing a large area now abandoned or occupied by the railway and pipeline construction debris, the new green area will contribute to improving biodiversity and will act as a "natural" buffer zone for protecting the qanat.

For Action C.5 "*Finding a balance between long-term conservation and short-term needs*", we propose one action:

- Development of a water management plan for the No-Ras qanat. This type of analysis must be conducted by the RWWTC, RAJO and the local Municipality in collaboration with local farmers, representing also the final beneficiaries. A good reference for drafting an efficient plan are the State of Conservation Reports [69,80], issued by ICHHTO and UNESCO for the current qanat included in the World Heritage List [81]. This could be integrated with the documentation from the qanat systems of Kashan and Gonabad (both in Iran), which are part of the GIAHS network [82,83].

### 3.3.2. Group of Activities 2

The recommendations were drafted to match the Action criteria A.5 and B.6 of the ICOMOS-IFLA guidelines. To meet them, we designed a specific action:

- Creation of an association/council for managing the No-Ras qanat. Following the example of "The Persian Qanat" UNESCO WHS [81], we proposed the creation of a Council of the No-Ras Qanat. This body is crucial for the daily management of the qanat (while the ICHHTO manages then at the national and regional level, see above), also acting as a bridge between the local, national and regional authorities and the community, including the farmers. Furthermore, the creation of a Council represents the first stage of ATN named Enrolment [66], which is essential to strengthening the network of actors. According to the national law (Articles 17 and 106) and other reports [81] and research [66], the council structure should be composed of 5–7 members and can be "*headed by the qanat council manager (mirab) and comprising the water clock operator (kayyal), the accountant (hesabdar), together with other qanat workers generally termed moqannis and another specialists such as the bucket operator and the windglass operator*". We suggest following this example, also including the representative of the local farmers and the municipality of Chavan and Tabriz to increase inclusiveness in decision-making.

In designing the role of the Council of the No-Ras Qanat, we also took into account the ICOMOS recommendation to "The Persian Qanat" UNESCO WHS. Therefore, the council will be responsible for: 1. developing a management plan, also including monitoring strategies; 2. implementing monitoring, conservation and reconstruction activities at the qanat in collaboration with ICHHTO, RWWTC and RAJO; 3. developing and implementing a risk preparedness plan with the support of external consultants, also meeting international standards, such as the UNESCO Disaster Risk Management [84] or the Sendai Framework [85]; 4. organizing training programs for qanat employees and new candidates; 5. managing water distribution for daily use and irrigation purposes in collaboration with the municipality and the local farmers.

### 3.3.3. Group of Activities 3

The recommendations proposed for group of activities 3 match the Action criteria A.6, B.4 and C.2 of the ICOMOS-IFLA guidelines. Tailor-made activities were designed to meet the requests.

For Actions A.6 "*Recognizing local populations as knowledge holders*" and C.2 "*Recognizing key stakeholders for rural landscapes, including rural inhabitants*", we propose the following action:

- Training courses for moqannis and professional figures working in the qanat. To this end, the Council of the No-Ras Qanat will evaluate be necessary for hiring new moqannis or other specific jobs related to the conservation and management of the qanat. If so, training sessions supported by the Municipality (for the economic support and the bureaucratic aspects) and the local university (for the details about the qanat engineering and heritage aspects) will be organized.

For Action B.4 "*Consider economic, social and environmental values*", we proposed the following action:

- Investing in non-invasive sustainable technologies to improve qanat efficiency. The importance of integrating modern non-invasive and sustainable technologies in ancient structures and systems, such as the qanats, has been recently emphasized by the European Union by promoting "Research and Innovation Actions" within the Horizon Europe program for reusing and improving traditional heritage crafts and systems.[4] The technological improvements that could be applied to the qanats are many and, here, we provide some examples. One of them is satellite imagery, a valuable and user-friendly tool for monitoring the state of conservation of the wells when access to them is prevented (i.e., during conflicts, natural disasters or pandemics) [61,86,87]. The

growing availability of open access imagery makes them more accessible and usable for everyone. Moreover, numerous researchers developed methodologies to identify and document endangered qanats [49,88,89]. Another example regards internal damage and stability that may be evaluated and mitigated through the construction of low-weight structures, spread and pile foundations or retrofitting (e.g., load decreasing, load-bearing columns and walls and load-diverting arches) [90,91]. One final solution is to keep the secondary road unpaved to avoid further spread of tarmac, using bio-enzyme products, such as Perma-Zyme, Terra-Zyme, or Fujibeton, which guarantee soil stability and the long-term durability of the unpaved road. These are economic and environmentally friendly solutions that have been extensively employed in countries, such as India [92], Egypt [86] and Australia [87]. The improvement in the current road plan will allow for a faster and simpler connection to the center of Chavan with Tabriz and the neighborhood village.

### 3.3.4. Group of Activities 4

The recommendations proposed for group of activities 4, "lack of awareness", match the Action criteria C.4, D.1 and D.2 of the ICOMOS-IFLA guidelines [33]. Tailor-made activities were designed to meet the requests.

For Action C.4, "*Considering interconnections between landscapes*", we propose the following action:

- Organizing visits to the qanat for children, families, schools, etc. The involvement of the local population in conservation, management and reuse activities to raise awareness on heritage and foster interconnection between urban and rural landscapes is widely recognized [88,89]. In our case, we suggest that the new-born Council of the No-Ras should work together with the different actors, including the local university, farmers and the municipality of Chavan, to organize visits to the qanat and events targeting schools and families and other members of the Chavan and Tabriz community. This activity also matches action D.2.

For Actions D.1, "*Communicating awareness*" and D.2 "*Increasing awareness*", we proposed the following actions:

- Developing signals and panels. To ensure understanding and use of the qanat by the local community, the Council of the No-Ras should collaborate with the University of Tabriz to design information panels about the history of the qanat and its benefits for agriculture, livestock, drinking water and the sustainability of qanats, compared to modern water systems, such as dams or hydro-pumping machines. This type of activity is linked to the transversal ones in response to Action B.3 in group of activities 1.
- Organizing conferences and meetings. In addition to the visits to the qanat expected for meeting the requirements of Action C.4, the council of the No-Ras qanat will work together with the local university and the municipalities of Chavan and Tabriz to organize conferences and other public events. These types of activities will serve for communicating and raising awareness on the historical, cultural and environmental value as well as the economic benefits of the qanat.
- Creating webpages and other online tools. Over the last few decades, the use of social media and websites to communicate the importance of cultural and natural heritage has jumped to the top of the agendas of many institutions and projects [93,94]. These media are particularly effective for sharing content with children and teenagers, meaning future generations that will be in charge of the conservation, management and use of the qanat. Therefore, the Council of the No-Ras qanat should hire a social media and website developer and manager to create the No-Ras qanat website and social media accounts. These will also be relevant for advertising other activities connected to group of activities 4, such as conferences, events and visits to the qanat, as well as group of activities 3, such as training and job offers.

3.3.5. Group of Activities 5

The recommendations proposed for Group of activities 5, "*lack of understanding of the qanat as heritage*", match the Action criteria A.4, A.7 and D.3 of the ICOMOS-IFLA document (2017). Tailor-made activities were designed to meet the requests.

For Actions A.4, "*Inventory and catalogue rural landscapes at all scales*" and A.7 "*Promoting cooperation for research*", we proposed the following activity:

- Scientific research on the history of the qanat. Researchers from the local university will coordinate a historical analysis on the No-Ras qanat. This multidisciplinary activity encompasses a number of different approaches for reconstructing the development of the qanat and its landscape. These include the use of historical archival documents about territorial management (e.g., cadasters or military maps) [95–97], spaceborne and airborne historical and contemporary imagery of the area [36,98], archaeological and geoarchaeological survey (including palaeobotanic analysis) [99–101] and community-based documentation and mapping [102–104].

For Actions D.3, "*Supporting shared learning, training and research*", we propose the following activity:

- Interviews to collect oral history. This type of activity is tightly related to the above-mentioned approaches. It can enrich the quality and quantity of information collected for the historical analysis of the No-Ras qanat, as well as strengthening the sense of belonging towards the place by local people [105]. This type of research is generally conducted by university researchers targeting local communities (mostly elders) as well as members of the council of the No-Ras qanat and local farmers. These stories may be shared with the rest of the community through events, conferences as well as online tools (see Group of activities 4).

**4. Discussion**

In this section, we discuss the potential limits and problems of each activity proposed to meet the recommendations and mitigate the issues illustrated by Branduini et al. [36]. Moreover, we propose to read the changes proposed in the light of "morality of water management", introduced by Ertsen [20].

*4.1. Potential Limits and Counteractions*

For the first issue, "Mitigate water shortage", the assessment of costs and benefits of the current hydraulic infrastructures may result in the possibility of a balance or positive feedback between the two. In this case, the conservation and reuse of the No-Ras qanat envisaged in this paper will improve the current situation, acting as an additional source of water supply. The implementation, conservation and enhancement activities will instead require a substantial amount of funding. If the Iranian government or the local institutions are unable to support them with through ad hoc funding, one possible option would be to apply for specific funds from the Horizon Europe Clusters 2—Culture Creativity and Inclusive Society (for the heritage aspect), Cluster 5—Climate, Energy and Mobility (for climate related issues), Cluster 6—Food, Natural Resources, Agriculture and Environment (for food), many of which have yet to come out. These kinds of calls embrace wide topics and aggregate a large number of stakeholders. This solution also applies to the development of the water protocol, which would be part of the implementation phase.

As for "Mis-management and/or lack of documentation" (Issue 2), some members of the local community or other actors identified in this study may not be willing to join the council. Therefore, to consider the council sufficiently represented, it would be necessary to have at least a qanat council manager (mirab), the water clock operator (kayyal) and the accountant (hesabdar), together with one representative from the local farmers and the municipality of Chavan. Other figures from national bodies, such as MCHT, RWWTC and RAJO, may not directly be part of the council or covering operational roles,

while it is essential for guaranteeing the council effectiveness to include members of the local community.

The activities related to "Loss of technical skills" (issue 3) may encounter the following types of problems: 1. lack of funding for organizing training and purchasing non-invasive sustainable technologies; 2. low participation of locals in training courses for moqannis and professional figures working in the qanat. In the first case, if the Iranian Government or local institutions are unable to provide funding for these activities, economic support can be sought in the Horizon Europe funds, in particular Clusters 2, 5 or 6 (see above). In addition to these lines of funds, the training can also be organized within the framework of capacity building projects, such as the EU-funded Erasmus+ Capacity Building KA1–3, involving not only local actors but also national bodies, private companies and universities. The second problem could be solved by extending the call for training also outside the borders of the region, thus, attracting people from other parts of the country.

As for issues 1 and 3, the activities related to issue 4 (lack of awareness) also present mainly potential economic problems. For example, economic support is crucial for developing signals and panels, creating webpages and other online tools, as well as for organizing conferences. For all these activities, we propose the same solution already identified in the cases of issues 1 and 3, to apply for the Clusters 2, 5 or 6 (see above) of the Horizon Europe programs.

In the event of low attendance at conferences and meetings, invitations can be extended to national and international guests in order to give greater prominence to this project. The same goes for visits to the qanat for children, families and schools. If local participation does not meet the expectations, schools and families from other regions of the country will be invited.

Problems related to the last issue may mostly arise due to the difficulties in conducting interviews for collecting oral history among the locals. One possible solution to solve the issue could be that of gathering only available stories from publications or the web.

### 4.2. Qanat as Moral Heritage

The solution proposed for the No-Ras qanat and the issues identified for its conservation and reuse also make it necessary to consider the relationship between the old infrastructure of the qanat and the people living in its premises, according to what the authors of [20] described as "moral landscape". The changes proposed are not neutral but will rather affect people that, in some cases, do not show direct personal or cultural connection with the qanats and their history. The social background may then divide the community between those looking at the project as something "bad" or "wrong" that will cause substantial damage, at least in the short term, and those who instead consider it "good" or "useful" and that it will enhance the area, reconnect the population to a partially forgotten past, also providing clean water in a sustainable way. Branduini et al., demonstrated that this dichotomy is partially due to the lack of awareness towards the history and benefits that this building may bring [36]. That is why Group of activities 4 and 5 are critical to allow the project to be fully understood by the entire community. In Ertsen's words, "*Relations need continuous confirmation, reconstruction and adaptation; maintaining stability is hard work for all human agents, precisely because other human agents and their non-human colleagues strike back*" [20].

### 5. Conclusions

In this article, we applied the guidelines from ICOMOS-IFLA on rural landscapes [33] to preserve, manage and reuse a historical hydraulic system, in particular, a water irrigation system, such as qanats and similar hydraulic structures, which are attested from the Mediterranean basin (e.g., Spain, Morocco, Italy, Libya, Egypt, Syria, etc.) to the Middle and Far East (e.g., Oman, Yemen, Iran, Armenia, Afghanistan and China), both in urban and rural areas. For each problem and recommendation that emerged in Branduini et al., several actions were also proposed based on the ICOMOS-IFLA guideline criteria [36].

In the discussion, we critically evaluated the potential problems affecting each activity proposed and provided possible solutions.

The testing of this methodology will allow for replication on any type of traditional rural landscape and it will:

1.  Raise awareness among communities at any level on the preservation of their past and cultural origins, thus, meeting the requests of SDG 11.4.
2.  Create a new sustainable economic resource passed down from the past to the present through interactive activities, thus, contributing to reach SDG 4 (lifelong learning).
3.  To preserve and reuse an environmentally friendly system that could, at least partially, replace the existing polluting solutions (i.e., dam and other modern infrastructures). This last point is nowadays at the top of the agenda of numerous national and international institutions and it matches SDG 6 (Clean Water and Sanitation) and SDG13 (Climate action).

In the specific case of the No-Ras qanat, the current proposal should be ideally used by the RWWTC, RAJO and the municipality in collaboration with the other actors and beneficiaries identified to draft a masterplan encompassing the different groups of activities and hypothesizing costs and potential funding opportunities. This will bring multiple benefits both for the urban community of Chavan in terms of urban layout rationalization, increases in the economic value of buildings and other public and private assets, attraction of tourists visiting the qanat as well as researchers and farmers visiting the Centre for Organic farming in order to export its methodologies and approaches.

However, more than anything, our hope is that this project will have a positive impact on both the cities and citizens of Chavan and Tabriz in raising awareness on this important Iranian traditional heritage.

**Author Contributions:** Conceptualization, F.Z. (Federico Zaina), P.B. and F.Z. (Fereshteh Zavvari); methodology, F.Z. (Federico Zaina), P.B. and F.Z. (Fereshteh Zavvari); analysis, F.Z. (Fereshteh Zavvari) resources, F.Z. (Fereshteh Zavvari); data curation, F.Z. (Federico Zaina); writing—original draft preparation, F.Z. (Federico Zaina); writing—review and editing, P.B.; visualization, F.Z. (Federico Zaina) and F.Z. (Fereshteh Zavvari). All authors have read and agreed to the published version of the manuscript.

**Funding:** This research received no external funding.

**Informed Consent Statement:** This paper, builds on a previous research [36], for which informed consent was already obtained from all the subjects involved.

**Data Availability Statement:** The data that support the findings of this study are available from the corresponding author upon reasonable request.

**Acknowledgments:** The authors wish to acknowledge the manager of the Fath Abad Garden and the moqanni of the No-Ras qanat for their help and support during the fieldwork and the extensive information provided, including photos and videos.

**Conflicts of Interest:** The authors declare no conflict of interest.

## Notes

1   https://ec.europa.eu/info/funding-tenders/find-funding/eu-funding-programmes/horizon-europe_en (accessed on 3 September 2022).
2   https://whc.unesco.org/en/list/?search=qanat&order=country (accessed on 3 September 2022).
3   https://www.fao.org/giahs/giahsaroundtheworld/designated-sites/asia-and-the-pacific/en/ (accessed on 3 September 2022).
4   https://ec.europa.eu/info/funding-tenders/opportunities/portal/screen/opportunities/topic-details/horizon-cl2-2022-heritage-01-04;callCode=null;freeTextSearchKeyword=traditional%20;matchWholeText=true;typeCodes=1,2,8;statusCodes=31094501,31094502,31094503;programmePeriod=null;programCcm2Id=null;programDivisionCode=null;focusAreaCode=null;destination=null;mission=null;geographicalZonesCode=null;programmeDivisionProspect=null;startDateLte=null;startDateGte=null;crossCuttingPriorityCode=null;cpvCode=null;performanceOfDelivery=null;sortQuery=sortStatus;orderBy=asc;onlyTenders=false;topicListKey=topicSearchTablePageState (accessed on 3 September 2022).

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
