# Peer review of "Applying ICOMOS-IFLA Principles for the Conservation, Management and Reuse of a Historical Hydraulic System: The No-Ras Qanat in North-Western Iran"

_heritage, doi:10.3390/heritage5040163_

Round 1

Reviewer 1 Report

thank you very much for the opportunity to know in detail your proposal for the enhancement of the hydraulic heritage in a specific case in Iran. I would like to make some suggestions that will help to improve your article in its final drafting. The first is that you should not assume that everyone knows the word Qanat, a Farsi word that might need an initial explanation and its translation in English and Arabic. On the other hand, I think you should explain, even briefly, the historical importance of the No-Ras qanat. Why is it important? Finally, although you point out some risks of the enhancement proposal you make, it is not clear whether this is merely a theoretical proposal or is already being implemented. What are the main threats to this proposal being carried out? 

Author Response

We wish to thank the reviewer for the useful comments. We tried to meet her/his requests by improving several sections as follows (all replies in red and italics):

thank you very much for the opportunity to know in detail your proposal for the enhancement of the hydraulic heritage in a specific case in Iran. I would like to make some suggestions that will help to improve your article in its final drafting. The first is that you should not assume that everyone knows the word Qanat, a Farsi word that might need an initial explanation and its translation in English and Arabic.

We wish to thank the reviewer for this comment. We added a short explanation in section 1.3 on the meaning of the word also mentioning the numerous variants and improving the reference on this topic. We also provided a concise definition for those who are not familiar with this hydraulic system.

On the other hand, I think you should explain, even briefly, the historical importance of the No-Ras qanat. Why is it important?

Thanks for this. We added in section 2.2 a short description of the historical relevance and the current importance and value that people give to this historical hydraulic system. Even in this case, we expanded the bibliography.

Finally, although you point out some risks of the enhancement proposal you make, it is not clear whether this is merely a theoretical proposal or is already being implemented. What are the main threats to this proposal being carried out? 

Thanks for this comment. We agreed that a clarification is necessary for the fact that this is a proposal in course of being approved (in section 3 and 3.1). Regarding the threats, we presume that the reviewer refers to the current threats to the qanat, not the potential threats of the project (to which the entire section 4.1 has been dedicated). Therefore, we expanded section 2.2 providing more details about the current threats to the qanat.

Reviewer 2 Report

thankyou for this wonderful body of work; the historiography was sound, the methods & aims were clear and the approach very useful in terms of findings. Great images, clear captions. I felt that Figure 7 offered a very long-term sustainable plan that other countries/water systems would benefit from, and  the authors could drive this point further in the conclusions...

the group activities will be a great training experience for university students, as well as provide economic & social benefits for the local community and protect the environment... the ideas expressed in this article serve to build capacity within the heritage sector

Author Response

Thank you for this wonderful body of work; the historiography was sound, the methods & aims were clear and the approach very useful in terms of findings. Great images, clear captions. I felt that Figure 7 offered a very long-term sustainable plan that other countries/water systems would benefit from, and  the authors could drive this point further in the conclusions...

the group activities will be a great training experience for university students, as well as provide economic & social benefits for the local community and protect the environment... the ideas expressed in this article serve to build capacity within the heritage sector.

We are so happy to have received such a positive review. On behalf of the rest of the authors, I wish to thank a lot the reviewer.

Reviewer 3 Report

The paper deals with a very interesting topic closely linked to the current debate on the conservation and management of the historical landscape, and of the rural one in particular, with a focus on hydraulic infrastructures.

It is well structured and clearly illustrates both the content and the research methodology: the authors deserve sincere compliments for their work. The proposed case study is also of absolute interest and deserves further development.

There are no substantial additions or corrections but only minor indications:

- lines 15-16: the sentence must be corrected (not only as “an” element “of” the past);

- use of “th” to indicate the centuries: sometimes the characters have the same size as the others, other times they are superscripts;

- use of we/our (lines 19, 55, 60, 157, 170, 177 ...): the use of personal pronouns is not recommended in scientific papers; impersonal speech is recommended instead.

A final suggestion concerns a possible comparison with similar case-studies: see the researches of the Land of Nineveh Archaeological Project (directed by Daniele Morandi Bonacossi). In particular see Morandi Bonacossi, D. 2018 - Water for Nineveh. The Nineveh Irrigation System in the Regional Context of the 'Assyrian Triangle': A First Geoarchaeological Assessment, in H. Kühne (ed.), Water for Assyria, Studia Chaburensia 7, Harrassowitz, Wiesbaden.

Author Response

We wish to thank the reviewer for the useful comments. We tried to meet her/his requests by improving several sections as follows (all replies in red and italics):

The paper deals with a very interesting topic closely linked to the current debate on the conservation and management of the historical landscape, and of the rural one in particular, with a focus on hydraulic infrastructures.

It is well structured and clearly illustrates both the content and the research methodology: the authors deserve sincere compliments for their work. The proposed case study is also of absolute interest and deserves further development.

There are no substantial additions or corrections but only minor indications:

- lines 15-16: the sentence must be corrected (not only as “an” element “of” the past);

Thanks for this comment. We have corrected the text accordingly.

- use of “th” to indicate the centuries: sometimes the characters have the same size as the others, other times they are superscripts;

We checked throughout the text and uniformed all the characters size.

- use of we/our (lines 19, 55, 60, 157, 170, 177 ...): the use of personal pronouns is not recommended in scientific papers; impersonal speech is recommended instead.

Thanks for this comment. Whenever possible, we have corrected the text accordingly.

A final suggestion concerns a possible comparison with similar case-studies: see the researches of the Land of Nineveh Archaeological Project (directed by Daniele Morandi Bonacossi). In particular see Morandi Bonacossi, D. 2018 - Water for Nineveh. The Nineveh Irrigation System in the Regional Context of the 'Assyrian Triangle': A First Geoarchaeological Assessment, in H. Kühne (ed.), Water for Assyria, Studia Chaburensia 7, Harrassowitz, Wiesbaden.

Thanks for this comment. We have included this and other publications as useful parallels.

Reviewer 4 Report

Review of heritage-1940917-peer-review-v1

The abstract needs to be rewritten as it presumes too much pre-knowledge and does not adequately set the scene for a heritage specialist who is not aware of the systems. It should also set ou the key findings of the paper. Also, abstracts should not contain references.

Line 34 f “ Moreover, the first strategies of water manipulation for irrigation purposes, naviga-tion and delimitation of borders, emerged as early as the Bronze Age in different regions 36 of the world [3,4]” These references are inadequate for the ambit claim made in the sentence

Line 39  “As stressed by [5] water”  who is [5]?  this is careless manuscript preparation.
the same applies to “discussed by [24].” In line 163. And also line 198

Line 55            In this context it would be good for the authors to also reference the use if ‘cultural water’ to several Indigenous Australian communities.

Line 61            “historical hydraulic landscape (hereafter HHS).”  Why is historical hydraulic landscape abbreviated to HHS and not HHL?

Line 77            UNESCO, ICOMOS and IFLA as, at least ICOMOS and IFLA must be fully spelled out before the acronym is used. UNESCO is universal and does not need to be explained

Line 115ff  thus is where the authors’ prior work (ref 1) needs to be added in

Line 119  you need to very briefly describe what a qanat is and how it works. You cannot assume that the readers know this

Line 202  “For the abbreviations see Section 3.1.”  No, you need to provide them as a footer to the table

Line 212  What is §3?

Section 2.2. is not  a methodology at all, but a list of criteria and actions from a previous paper. There is no justification given for any of this. The present paper has to stand on its own and cannot rely on readers chasing an obscure conference paper. The methodology sections needs to fully reconceptualised and rewritten. If I were a reader, and not a reviewer, I would stop right here and discard the paper as irrelevant top me…if I cannot understand the methodology as it has not been presented, I will a priori disregard any findings and will most certainly not cite the paper. And I am not the only one in ‘this camp.’ So it behoves the authors to fix this section of they want to be taken seriously.

The remaining bulk of the paper is fine and makes sense to me. The only serious  concern I have is that the content diverges very much from the concept of heritage into the area of urban planning and thus the article would possibly be much more suited to a submission to MDPI Land as opposed to MDPI Heritage.

Minor issues

The paper needs to undergo a thorough edit by a professional, native-English speaking editor for sentence structure and expression. As it stans it contains numerous typographical errors, wrong use of terms and awkward expressions. This includes the legends of images (eg, Fig 1 what is ‘urban spatial”?)

Author Response

We wish to thank the reviewer for the useful comments. We tried to meet her/his requests by improving several sections as follows (all replies in red and italics):

The abstract needs to be rewritten as it presumes too much pre-knowledge and does not adequately set the scene for a heritage specialist who is not aware of the systems. It should also set out the key findings of the paper. Also, abstracts should not contain references.

Thanks for this comment. We totally agree with the reviewer and we changed the abstract accordingly.

Line 34 f “ Moreover, the first strategies of water manipulation for irrigation purposes, navigation and delimitation of borders, emerged as early as the Bronze Age in different regions 36 of the world [3,4]” These references are inadequate for the ambit claim made in the sentence.

We thank the reviewer for these comments. We improved the body of reference in order to meet her/his request. If these are still insufficient, we would be happy to receive any suggestions.

Line 39  “As stressed by [5] water”  who is [5]?  this is careless manuscript preparation.
the same applies to “discussed by [24].” In line 163. And also line 198

I’m sorry, but we don’t understand this point. This is not careless manuscript preparation on the contrary it is exactly the way texts must be quoted according to the MDPI system. In order to be sure that we were applying it correctly, we checked both the guidelines and other papers as an example and both confirmed that our use of the reference is correct. To know who is [5] or [24] the reader just needs to check in the reference.

Line 55 In this context it would be good for the authors to also reference the use if ‘cultural water’ to several Indigenous Australian communities.

Thanks for this useful suggestion. We mentioned this particular case along with the other two also adding specific references. If the reviewer has more references to suggest we would be happy to add them.

Line 61 “historical hydraulic landscape (hereafter HHS).”  Why is historical hydraulic landscape abbreviated to HHS and not HHL?

Thanks for this comment. We corrected and uniformed all the abbreviations as HHS (i.e. historical hydraulic systems).

Line 77  UNESCO, ICOMOS and IFLA as, at least ICOMOS and IFLA must be fully spelled out before the acronym is used. UNESCO is universal and does not need to be explained

We agree with the reviewer and we fully spelt all the acronyms.

Line 115ff  thus is where the authors’ prior work (ref 1) needs to be added in

Thanks for this comment. Done.

Line 119  you need to very briefly describe what a qanat is and how it works. You cannot assume that the readers know this

We wish to thank the reviewer for this comment. We added a short explanation in section 1.3 on the meaning of the word also mentioning the numerous variants and improving the reference on this topic. We also provided a concise definition for those who are not familiar with this hydraulic system.

Line 202  “For the abbreviations see Section 3.1.”  No, you need to provide them as a footer to the table

Thanks for this comment. Done.

Line 212  What is §3?

Corrected

Section 2.2. is not a methodology at all, but a list of criteria and actions from a previous paper. There is no justification given for any of this. The present paper has to stand on its own and cannot rely on readers chasing an obscure conference paper. The methodology sections needs to fully reconceptualised and rewritten. If I were a reader, and not a reviewer, I would stop right here and discard the paper as irrelevant top me…if I cannot understand the methodology as it has not been presented, I will a priori disregard any findings and will most certainly not cite the paper. And I am not the only one in ‘this camp.’ So it behoves the authors to fix this section of they want to be taken seriously.

We thank the reviewer for this important comment. We agree that the description of the methodology could make it a bit difficult to fully understand the paper (although no other reviewer has emphasized this point, while reviewers 1 and 2 praised the clarity of the whole paper) since it builds upon previous research (which is not by the way from any obscure conference paper but from one that is open access and easily accessible from this link http://ocs.editorial.upv.es/index.php/arqueologica20/arqueologica9/paper/view/12102). Therefore, we substantially improved this part allowing the reader to understand how the list of criteria and actions emerged. We believe that now the text is clearer and standing alone as requested by the reviewer. We also don’t think that it could be added more because the risk is that of an excessive overlapping with the previous research leading to plagiarism.

We believe, and we are not the only one, that if a paper builds on previous research (which the case of many papers) the authors don’t need to provide all the details of that research, but only enough information for guaranteeing its clarity.

The remaining bulk of the paper is fine and makes sense to me. The only serious concern I have is that the content diverges very much from the concept of heritage into the area of urban planning and thus the article would possibly be much more suited to a submission to MDPI Land as opposed to MDPI Heritage.

We thank the reviewer for this important point. However, we believe that our subject perfectly fits into the area of urban planning. The qanat literally runs inside a village and for a long time provided water to the city of Tabriz. In the case of Chavan town, our proposal (positively) impacts its urban layout not only physically but also in terms of stakeholders’ involvement. This point has been touched on in multiple passages of the paper, but to stress this more and meet the reviewer’s request we added this point elsewhere.

Although we believe that similar research might match the requests also of a journal like MDPI Land the stronger focus on heritage and the relation of this heritage with an urban area (both the town of Chavan and Tabriz) makes it suitable for MDPI Heritage.

Minor issues

The paper needs to undergo a thorough edit by a professional, native-English speaking editor for sentence structure and expression. As it stans it contains numerous typographical errors, wrong use of terms and awkward expressions. This includes the legends of images (eg, Fig 1 what is ‘urban spatial”?)

We agree with the reviewer. An English native speaker has also gone through the paper including the figures.

Round 2

Reviewer 4 Report

In  my initial review I wrote:

Line 34 f “ Moreover, the first strategies of water manipulation for irrigation purposes, navigation and delimitation of borders, emerged as early as the Bronze Age in different regions 36 of the world [3,4]” These references are inadequate for the ambit claim made in the sentence.

The authors responded

We thank the reviewer for these comments. We improved the body of reference in order to meet her/his request. If these are still insufficient, we would be happy to receive any suggestions.

Second review comment

The replaced references are extremely generic [4] and not applicable as they refer to the historic period [3]. This still needs to be fixed. It is hardly the role of myself as the reviewer to go and find references for the authors.

====================================

In  my initial review I wrote:

Line 39  “As stressed by [5] water”  who is [5]?  this is careless manuscript preparation.
the same applies to “discussed by [24].” In line 163. And also line 198

The authors responded

I’m sorry, but we don’t understand this point. This is not careless manuscript preparation on the contrary it is exactly the way texts must be quoted according to the MDPI system. In order to be sure that we were applying it correctly, we checked both the guidelines and other papers as an example and both confirmed that our use of the reference is correct. To know who is [5] or [24] the reader just needs to check in the reference.

Second review comment

I am quite confused by the seemingly unprofessional approach of the authors who clearly do not comprehend the basics of academic writing (but that may be a translation/cultural issue)…This is the way it should be done:

“As stressed by Hein et al, water can be a multifaceted tool in support of humans [4],

====================================

In  my initial review I wrote:

The remaining bulk of the paper is fine and makes sense to me. The only serious concern I have is that the content diverges very much from the concept of heritage into the area of urban planning and thus the article would possibly be much more suited to a submission to MDPI Land as opposed to MDPI Heritage.

The authors responded

We thank the reviewer for this important point. However, we believe that our subject perfectly fits into the area of urban planning. The qanat literally runs inside a village and for a long time provided water to the city of Tabriz. In the case of Chavan town, our proposal (positively) impacts its urban layout not only physically but also in terms of stakeholders’ involvement. This point has been touched on in multiple passages of the paper, but to stress this more and meet the reviewer’s request we added this point elsewhere.

Although we believe that similar research might match the requests also of a journal like MDPI Land the stronger focus on heritage and the relation of this heritage with an urban area (both the town of Chavan and Tabriz) makes it suitable for MDPI Heritage.

Second review comment

The entire section 3 is, essentially, an urban planning document. For it to stand in MDPI Heritage the relevance the proposed actions to the cultural heritage of the town need to be  much  more developed. As it stands, the paper is, IMHO, an ill fit foe MDPI Heritage, but the authors can address and fix this in the manuscript. I have hoped that the revised version would have addressed this, but it did not.

Author Response

In my initial review I wrote:

Line 34 f “ Moreover, the first strategies of water manipulation for irrigation purposes, navigation and delimitation of borders, emerged as early as the Bronze Age in different regions 36 of the world [3,4]” These references are inadequate for the ambit claim made in the sentence.

The authors responded

We thank the reviewer for these comments. We improved the body of reference in order to meet her/his request. If these are still insufficient, we would be happy to receive any suggestions.

Second review comment

The replaced references are extremely generic [4] and not applicable as they refer to the historic period [3]. This still needs to be fixed. It is hardly the role of myself as the reviewer to go and find references for the authors.

Second response of the authors:

We have improved the bibliography by adding 10 specific publications for different areas of the world including the Near East, South America, India, China and Europe

====================================

In  my initial review I wrote:

Line 39  “As stressed by [5] water”  who is [5]?  this is careless manuscript preparation.
the same applies to “discussed by [24].” In line 163. And also line 198

The authors responded

I’m sorry, but we don’t understand this point. This is not careless manuscript preparation on the contrary it is exactly the way texts must be quoted according to the MDPI system. In order to be sure that we were applying it correctly, we checked both the guidelines and other papers as an example and both confirmed that our use of the reference is correct. To know who is [5] or [24] the reader just needs to check in the reference.

Second review comment

I am quite confused by the seemingly unprofessional approach of the authors who clearly do not comprehend the basics of academic writing (but that may be a translation/cultural issue)…This is the way it should be done:

“As stressed by Hein et al, water can be a multifaceted tool in support of humans [4],

Second response of the authors:

We have changed the passages according to the reviewer’s suggestions.

====================================

In my initial review I wrote:

The remaining bulk of the paper is fine and makes sense to me. The only serious concern I have is that the content diverges very much from the concept of heritage into the area of urban planning and thus the article would possibly be much more suited to a submission to MDPI Land as opposed to MDPI Heritage.

The authors responded

We thank the reviewer for this important point. However, we believe that our subject perfectly fits into the area of urban planning. The qanat literally runs inside a village and for a long time provided water to the city of Tabriz. In the case of Chavan town, our proposal (positively) impacts its urban layout not only physically but also in terms of stakeholders’ involvement. This point has been touched on in multiple passages of the paper, but to stress this more and meet the reviewer’s request we added this point elsewhere.

Although we believe that similar research might match the requests also of a journal like MDPI Land the stronger focus on heritage and the relation of this heritage with an urban area (both the town of Chavan and Tabriz) makes it suitable for MDPI Heritage.

Second review comment

The entire section 3 is, essentially, an urban planning document. For it to stand in MDPI Heritage the relevance the proposed actions to the cultural heritage of the town need to be  much  more developed. As it stands, the paper is, IMHO, an ill fit foe MDPI Heritage, but the authors can address and fix this in the manuscript. I have hoped that the revised version would have addressed this, but it did not.

Second response of the authors:

We have more extensively explained how the proposed actions will be relevant to the cultural heritage of both Chavan village and Tabriz city. To strengthen our arguments, we also included dedicated new references and best practices. The reviewer can find it in red not only in section 3 but also in the conclusion.